# Assessing the Efficacy of the 3R (Reframe, Reprioritize, and Reform) Communication Model to Increase HPV Vaccinations Acceptance in Ghana: Community-Based Intervention

**DOI:** 10.3390/vaccines11050890

**Published:** 2023-04-24

**Authors:** Matthew Asare, Peter Agyei-Baffour, Adofo Koranteng, Mary E. Commeh, Emmanuel Sarfo Fosu, Anjelica Elizondo, Rodney X. Sturdivant

**Affiliations:** 1Department of Public Health, Robbins College of Health and Human Sciences, Baylor University, Waco, TX 76798, USA; anjelica_elizondo@baylor.edu; 2School of Public Health, Kwame Nkrumah University of Science and Technology, Kumasi P.O. Box UPO-1279, Ghana; pagyei-baffour.chs@knust.edu.gh (P.A.-B.); kadofo.chs@knust.edu.gh (A.K.); 3Non-Communicable Disease Control, Ghana Health Services, Accra P.O. Box MB-582, Ghana; mary.commeh@ghs.gov.gh; 4Department of Statistical Science, College of Arts & Sciences, Baylor University, Waco, TX 76798, USA; emmanuel_sarfofosu1@baylor.edu (E.S.F.); rodney_sturdivant@baylor.edu (R.X.S.)

**Keywords:** HPV vaccination, intervention, 3-R communication model, Ghanaian parents, adolescents

## Abstract

The study examined whether the 3R (reframe, prioritize, and reform) communication model intervention can impact parents’ and adolescents’ HPV vaccination acceptability. We used face-to-face methods to recruit participants from three local churches in the Ashanti Region of Ghana. Participants completed pre- and post-intervention assessments based on the validated Theory of Planned Behavior survey. We organized two face-to-face presentations for parents and adolescents separately for parents (*n* = 85) and adolescents (*n* = 85). Participants’ post-intervention vs. pre-intervention scores for attitude (mean = 35.46 ± SD = 5.46 vs. mean = 23.42 ± SD = 8.63), knowledge (M = 28.48 ± SD = 5.14 vs. M = 16.56 ± SD = 7.19), confidence (M = 8.96 ± SD = 3.43 vs. M = 6.17 ± SD = 2.84), and intention for vaccine acceptance (M = 4.73 ± SD = 1.78 vs. M = 3.29 ± SD = 1.87) increased significantly (*p* < 0.001). The intervention showed that for every one-unit increase in the participants’ self-confidence and attitude scores, the odds of the HPV vaccination acceptability increased by 22% (95% CI: 10–36) and 6% (95% CI: 0.1–12), respectively. Intention for vaccine acceptance, F (1167) = 6.89, and attitude toward vaccination, F (1167) = 19.87, were significantly higher among parents than adolescents (*p* < 0.001), after controlling for the baseline scores. These findings suggest that the intervention targeting parents’ and adolescents’ attitudes and knowledge has the potential to increase HPV vaccination acceptance in Ghana.

## 1. Introduction

Human papillomavirus (HPV) is the most common sexually transmitted disease in the world [1,2]. Incessant infection with oncogenic HPV is associated with cervical cancers and many vulvar, vaginal, penile, anal, oral, oropharyngeal, neck, and head cancers [3]. While malaria and HIV/AIDS have dominated the funding and attention of African government, nongovernmental, and philanthropic organizations, HPV-related cancers which are largely preventable through HPV vaccines or screening [4] are silently claiming the lives of Africans in great numbers [5]. In 2018, Africa had 190,000 HPV-related cancer cases and over 135,000 deaths [5]. In that same year, in Ghana, 14.47% (almost 4000) of new cancer cases were diagnosed, and the largely preventable HPV-related cancers were responsible for 19.73% (2600) of cancer deaths [6]. It is estimated that over nine million women aged 15 years and above in Ghana are at risk for cervical cancer [5]. Indeed, cervical cancer is the second leading cause of female cancer deaths in Ghana [5,7,8]. Unfortunately, there are no available data on HPV-related cancer among men in Ghana, but it is well documented that HPV infection is very common in sub-Saharan Africa [9], and over 135,000 women [5] needlessly die from preventable cancer each year. In reality, there is limited access to the screening and treatment for HPV-related cancers in low- and middle-income countries (LMICs), making the HPV vaccine the best hope for preventing these cancers in Ghana [10].

Evidence shows that human papillomavirus (HPV) vaccines (Cervarix^®^, Gardasil^®^, and Gardasil 9^®^) have contributed to the decline in HPV-related cancer cases and deaths [11,12,13,14]. World Health Organization (WHO) recommends two doses of vaccination for 9–14 year olds [11,15,16,17,18]. In 2013, Ghana pilot-tested HPV vaccination in collaboration with the Global Alliance for Vaccines and Immunizations (GAVI) in four districts [10,19,20]. GAVI offers subsidies for HPV vaccines in eligible LMICs, including Ghana. However, Ghana’s national immunization technical advisory group (NITAG) has yet to recommend an introduction of the HPV vaccine to its national immunization schedule [21]. Although the data on the HPV vaccination rate in Ghana are currently unavailable [22,23], it is well documented that low-income countries in the world, such as Ghana, have the lowest vaccination rate [5,24]. For instance, in high-income countries, 33.6% of females aged 10–20 years received the complete series of vaccines, compared with only 2.7% of females in LMICs [24].

Previous research suggests that logistical and attitudinal barriers contribute to low vaccinations in LMICs including in Ghana. Logistical factors include the cost of vaccination, lack of data on HPV vaccination, and unavailability of a widespread, systematic, publicly available HPV vaccination program [25,26]. Attitudinal or predisposing factors including lack of awareness or knowledge of the vaccine and its efficacy, stigma of vaccination, and religious objections are barriers to vaccination [27,28,29,30]. Parents’ perceptions about the HPV vaccine directly influence HPV vaccination uptake. However, our understanding of Ghanaian parents’ attitudes and behaviors toward HPV vaccination is limited. Additionally, many parents have an incomplete or inaccurate understanding of HPV vaccines which contributes to a general hesitancy to consider vaccination for their adolescents. The literature on these barriers further provides a need for an effective intervention to target changing attitudes and perceptions of parents and their unvaccinated adolescents.

Studies have shown that interventions designed to increase knowledge about the effectiveness of HPV vaccination are strongly associated with an increase in HPV vaccination uptake [31]. In a longitudinal cohort study, researchers following adolescents for 1 year after implementing an HPV awareness campaign found a significant increase in HPV vaccination among the cohort [32]. However, theoretical-based structured educational interventions to create awareness about HPV vaccination are understudied. We integrated a 3R (reframe, prioritize, and reform) communication model with the theory of planned behavior to promote HPV vaccination among Ghanaian parents and their unvaccinated adolescents.

### Theoretical Framework

The 3R communication model: Information on HPV vaccination can be maximally persuasive depending on how the information is framed [33]. HPV vaccination information can emphasize the benefits of taking action (i.e., a gain-framed appeal) or the costs of failing to take action (i.e., a loss-framed appeal) [33,34]. The taxonomy of gain–loss-framed appeals argues that gain-framed appeals are effective when people are considering a behavior that involves a relatively low risk of an unpleasant outcome (e.g., HPV vaccination prevents the onset of HPV-related cancers) [33,34,35]. On the other hand, loss-framed appeals are more effective when people are considering a behavior that they perceive involves some minimal risk of an unpleasant outcome (e.g., the risk for cervical HPV vaccination is possible pain) [33,34,35]. The 3R model provides a framework to simplify HPV vaccination messages. For instance, reframing, reprioritizing, and reforming messages have been shown to be effective to overcome stigma and religious objections associated with mammogram usage [36]. Second, several studies have pursued how the framing of a health message might affect people’s willingness to perform a particular behavior [37,38]. Loss-framed appeals have been successfully used to promote mammography and breast self-examination screening [39,40,41,42,43], as well as colorectal cancer screening [44,45]. With the proliferation of misinformation about vaccinations, several researchers have studied the effects of framed messages on perception, attitude, and intention toward vaccines [46,47,48].

The Theory of Planned Behavior (TPB), which posits that behavioral intention is the most proximal determinant of social behavior [49,50], can help explain intentions, attitudes, and perceived behavioral control for vaccination of parents and adolescents. We determined whether the expected positive impact of the intervention on the parents’ decision making for vaccination was explained by TPB constructs (behavioral intention, attitude, and behavioral control self-efficacy) and knowledge. Evidence of mediation will provide valuable insights into mechanisms underlying the intervention’s efficacy and may suggest ways to further enhance its ability to facilitate decision-making about HPV vaccination. Therefore, the purpose of this study was to determine whether the 3R (reframe, prioritize, and reform) communication model intervention can impact parents’ and adolescents’ knowledge, attitude, and perceptions regarding HPV vaccination acceptability.

## 2. Materials and Methods

Study design: We conducted a quasi-experimental (no control group) community-based intervention study. We recruited parents/caregivers (hereafter referred to as caregivers) and their 12–17 year old adolescents from three local churches in the Ashanti Region of Ghana.

Sample size determination: A sample size to achieve the primary aim of the study was projected to be 64 adolescents and 64 caregivers, calculated using the following criteria: effect size of 0.25; alpha of 0.05; power of 0.95; one group; two measurements; correction among representation measures of 0.5; and non-sphericity correction of 1, consistent with the literature [51]. Inputting those criteria in G*Power, a sample size of 54 adolescents and 54 caregivers was needed. To account for potential attrition, the sample size was increased by 20%; therefore, a total sample size of 64 was used in this study.

Participant Eligibility: A caregiver was broadly defined as an adult, older than 18 years old, who has the responsibility of taking care of an adolescent girl aged 12–17 years old. Caregivers could be a mother, a father, or legal guardian. Caregivers were included in the study if they met the following inclusion criteria: taking care of unvaccinated adolescents between ages of 12 and 17 years, able to give consent and assent for their adolescents, and willing to enroll their adolescent in the study. An adolescent was defined as a boy or girl between the age of 12 and 17 years old. Adolescents were included in the study if they met the following inclusion criteria: 12–17 years old, not vaccinated, and whose caregiver agreed to be part of the study.

Recruitment process: The study utilized a purposive sample and a snowball method to recruit the participants. A member of the research team contacted three different local church pastors and introduced the study to them. The study was announced at church service, and the congregations were encouraged to meet with the research team member who was waiting in the church lobby. After church service, the researcher met caregivers and their adolescents who were interested in the study, explained the study’s purpose, and scheduled a day for the intervention presentation. Participants were encouraged to introduce the study to other people who might be interested. Caregivers who expressed interest were screened for eligibility, and those eligible were invited to give verbal consent and additionally signed a parental permission form for their adolescents. Adolescents whose parents expressed interest were screened for eligibility, and a parental permission form was given to their caregivers for approval. Each participating adolescent gave verbal assent prior to participating in the study. Each participant received GHS 10 as an incentive for their participation.

### 2.1. Intervention Description

Intervention Delivery: We organized six intervention presentations on six different days: three presentations for the caregivers, and another three presentations organized separately for the adolescents. Each presentation was 1 h long covering the three modules, and it was a group face-to-face format. Each participant attended one presentation and thus completed all three modules. In each group, there were a minimum of 25 participants and a maximum of 30 participants. Following each factual presentation about the vaccination and the HPV virus, we used a loss-framed appeal throughout to guide all discussions.

Module 1 content covered reframing the conversations about the HPV vaccination as protection from cancer rather than sexually transmitted diseases. Specifically, in Module 1, the presentation covered topics such as facts and figures about HPV prevalence and incidence, causes of cervical cancer, mortality rates, and overall health implications of HPV infections. Examples of the loss-framed appeals we presented are as follows: There are many benefits you may not experience if you do not accept vaccination. If you decide not to vaccinate, you will not feel the peace of mind that comes with knowing that your child is protected, or you are protected. There are many problems or bad things (e.g., death or treatment of chemo and its side-effects) you may experience if you do not accept the vaccination. If you decide not to get a vaccination, you may feel more anxious because you may wonder if you are ill.

Module 2 content covered reprioritizing cervical cancer prevention as a healthcare priority for adolescents by creating awareness, building behavioral confidence, and reinforcing positive beliefs about HPV vaccination as a cancer prevention strategy. Specifically, the Module 2 presentation covered topics such as facts on HPV vaccination, e.g., efficacy, safety, dosage schedule, and recommendations. Examples of the reprioritizing message are as follows: By not vaccinating now because of the cost, the future expenses for cancer treatment will be high compared to the cost of vaccination and other possible costs including job and income loss, and even possible death that can be prevented. Research shows that adolescents who do not vaccinate have a decreased chance of preventing the virus. You can lose several potential health benefits by failing to vaccinate. Do not fail to take advantage of this opportunity because the window for maximum protection with the use of this vaccine is narrow; thus, the urgency for vaccination is critical and immediate.

Module 3 content covered reforming the belief system by teaching ways for the participants to confront the barriers (e.g., stigma and belief about promoting promiscuity) to HPV vaccination and helping them to uncover the logical flaws/or misinterpretation of HPV vaccinations. Specifically, in Module 3, the presentation explored and discussed myths and misconceptions about the HPV vaccines, belief systems about disease prevention, treatments and healings from traditional and religious points of view, and shared information about the HPV vaccine as a cancer preventative intervention. Examples of reforming messages used are as follows: Not vaccinating now will increase your chance of being infected with the virus. The fear of HPV vaccination pain is a legitimate concern, but the fear of the future tumor metastasizing (spreading) should outweigh the initial fear of pain.

### 2.2. Measures

Participants completed baseline and post-intervention assessments based on a validated Theory of Planned Behavior instrument [52]. Pre-intervention assessment: Each adolescent and caregiver individually completed a self-report questionnaire at the baseline. A hard copy of the questionnaire assessing the participants’ demographic information, knowledge, attitude, perceived behavioral control, and intention (attitude, perceived behavioral control, and intention items were the TPB constructs) for vaccine acceptance was hand-delivered to the participants to complete before the intervention. Post-intervention assessment: Soon after the intervention, each participant completed the questionnaire again. The primary outcome was acceptance for vaccination (defined as the likelihood of parents accepting HPV vaccine for their adolescent in the next month or the likelihood of adolescent accepting vaccination in the next month (one item). The secondary outcomes were changes in pre-intervention and post-intervention scores for knowledge about vaccination and cervical cancer (four items), positive attitude toward vaccinations (two items), negative attitude toward vaccinations (two items), and perceived behavioral control for vaccinations (two items), all measured on a five-point Likert scale.

### 2.3. Analyses

Descriptive statistics, such as frequencies and means, were used to analyze the demographic and other covariate data. A logistic regression model was used to analyze the associations between the independent variables and the dependent variable data. We used the paired-sample *t*-test to analyze the pre-test and post-test data and the ordinal logistic regression to compare the post-intervention score differences between the parents and adolescents while controlling for their baseline scores. A significant result was set a priori at a *p*-value < 0.05. The proportional odds assumption of the ordinal logistic model scale was tested. All data were analyzed using R software, version 4.2.2 [53].

## 3. Results

Proportional odds assumption: The results for the overall ordinal logistic regression model indicated that the proportional odds assumption was not violated. However, in the parents’ data, the proportional odds assumption was violated. We recoded the levels of the intention variable from a five-point Likert scale (extremely unlikely (1), unlikely (2), neutral (3), likely (4), and extremely likely (5)) to three categories (unlikely (1), neutral (2), and likely (3)), which improved the assumption results. After recoding, the conclusion drawn from the model did not change the original conclusion. However, the recorded variable enabled the model to meet the proportional odds assumptions.

Parents’ demographic characteristics: Table 1 shows the demographic distribution of the parents and the intervention effects on the demographic characteristics. A total of 170 participants including parents (*n* = 85) and adolescents (*n* = 85) provided evaluable data. The mean age of the parents and adolescents were mean = 45.89 ± SD = 12.15 and mean =14.35 ± SD = 1.70, respectively. Ninety-two percent of the parents and all the adolescents who participated in the study were females. The post-intervention results indicated a significant change in mean scores for knowledge about HPV vaccination for parents who had annual household income less than GHS 20,000 (~USD 2500), had no educational background, had less than a high school degree, and had an undergraduate degree, with increases by 16.78, 12.58, 11.77, and 12.40 points, respectively (*p* < 0.001). The post-intervention results also showed that parents with no educational background and those with income less than GHS 20,000 (~USD 2500) saw a significant change in their mean scores for their attitude toward HPV vaccination, their mean scores increased by 14.00 and 15.83 points, respectively (*p* < 0.001). Furthermore, male parents, as well as those who had never married, had annual household income less than GHS 20,000 (~USD 2500), and had no health insurance saw a significant change in the scores for their likelihood or intentions of allowing their adolescent to get vaccinated (*p* < 0.001). Scores of the intention to allow their adolescent to vaccinate for male parents increased by 3.14, for those were never married increased by 2.0, for those with income less than GHS 20,000 increased by 2.63, and for those with no health insurance increased by 2.61.

### 3.1. Main Intervention Effects

Comparing the two periods (baseline and post-intervention periods), we found an overall increase in post-intervention scores for both adolescents and parents. The results showed that there was a significant increase in participants’ post-intervention scores vs. pre-intervention scores for confidence (M = 8.96 ± SD = 3.43 vs. M = 6.17 ± SD = 2.84), knowledge (M = 28.48 ± SD = 5.14 vs. M = 16.56 ± SD = 7.19), intention for vaccination (M = 4.73 ± SD = 1.77 vs. M = 3.29 ± SD = 1.87), and positive attitude (M = 35.48 ± SD = 5.46 vs. M = 23.42 ± SD = 8.63), (*p* < 0.001). However, there was a significant decrease in post-intervention score for negative attitudes. Figure 1 shows the aggregated intervention scores for various behavioral outcomes. Table 2 shows changes in the pre-and post-intervention scores by individual group and the combined group.

### 3.2. Intervention Effect on Primary Outcome

Parents and adolescents: After controlling for the baseline assessment, we found that participants’ likelihood of HPV vaccination acceptance was associated with their post-intervention self-confidence scores (adjusted odds ratio (AOR) = 1.19, 95% CI: 1.08, 1.30). This indicates that, for every one-unit increase in the post-intervention self-confidence scores, the odds of participants’ intention for HPV vaccination acceptance increased by 19%. The participants’ likelihood of HPV vaccination acceptance was associated with their post-intervention positive attitude scores (AOR = 1.09, 95% CI: 1.03, 1.10). This indicates that, for every one-unit increase in the post-intervention positive attitude scores, the odds of participants’ intention for HPV vaccination acceptance increased by 9%.

Parents only: For every one-unit increase in the intervention score of parents’ confidence, the odds of a higher (vs. lower) likelihood of vaccine acceptance for their adolescents increased by 28% while holding baseline scores constant [AOR = 1.28 (1.09, 1.51)]. For every one unit increase in the intervention score of parents’ positive attitudes, the odds of a higher (vs. lower) likelihood of vaccine acceptance for their adolescent increased by 21% while holding baseline scores constant [AOR = 1.21 (1.03, 1.43)]. For every one-unit increase in the intervention score of parents’ negative attitudes, the odds of a lower (vs. high) likelihood of vaccine acceptance for adolescents increased by 15% while holding baseline scores constant [AOR = 1.15 (1.04, 1.28)].

Adolescents only: For every one-unit increase in the intervention score of adolescents’ confidence, the odds of a higher (vs. lower) likelihood of vaccine acceptance increased by 18% while holding baseline scores constant [AOR = 1.18 (1.04, 1.34)]. For every one-unit increase in the intervention scores of adolescents’ positive attitudes, the odds of a higher (vs. lower) likelihood of vaccine acceptance increased by 7% while holding baseline scores constant [AOR = 1.07 (1.00, 1.16)]. Table 3 shows the adjusted odds ratio of parent’s and adolescent’s confidence, positive attitude, knowledge, and negative attitude predicting their intention to vaccinate.

## 4. Discussion

This community-based intervention assessed whether the 3R communication model was an effective model in changing Ghanaian parents and their unvaccinated adolescents’ attitudes, knowledge, and intention for HPV vaccination acceptance. The first main takeaway of the intervention is that parents with varied educational backgrounds, income status, and insurance status benefited from the intervention. Parents with a less educational background gained significant knowledge about HPV vaccination, and they also developed positive attitudes toward HPV vaccination. Parents with a low income had a significant improvement in their intention to accept HPV vaccination and their attitude toward vaccination. Our findings support a previous study which concluded that structure–educational interventions have the potential to improve parental awareness, knowledge, and perceptions toward HPV and the acceptability of the vaccine [48].

A second major finding is the overall impact of the intervention on both parents and adolescents. The intervention showed significant effects on the adolescent and parent attitude toward HPV vaccination. Individuals’ positive attitudes toward a given behavior are associated with performing that behavior, whereas negative attitudes deter the performance of that behavior [54]. The post-intervention increase in participants’ scores for attitude indicates that participants had a favorable attitude toward vaccination, which could more likely lead to vaccination acceptance when given the opportunity [54]. Lack of knowledge about HPV vaccination is a known barrier to vaccination [27,28,29]. Our finding revealed a significant increase in knowledge among participants about vaccination. As more people become aware of the availability and effectiveness of the vaccination, the chances of getting vaccinated will increase [54]. Confidence in performing a behavior can lead to engaging in that behavior [55]. The participants’ score for confidence significantly increased after receiving the intervention, indicating that participants had strong confidence in accepting HPV vaccination.

Individuals’ behavioral intention is proximal to performing a behavior, and participants’ scores for intention for getting vaccination increased. Overall, the 3R communication model was effective in influencing the participants’ intentions for HPV vaccination acceptance. We found that participants’ likelihood of HPV vaccination acceptance was associated with their self-confidence. Our findings showed that, for every one-unit increase in the self-confidence scores, the participants’ likelihood of getting the HPV vaccination increased by 19%. The participants’ likelihood of HPV vaccination acceptance was associated with their positive attitude scores. For every one-unit increase in the post-intervention positive attitude scores, the participants’ likelihood of getting the HPV vaccination increased by 9%. Our findings concur with other studies suggesting that attitudes about the vaccine are the most proximal determinants of vaccine uptake and likely mediate some of the effects observed by the other variables [56,57]. Our study, in conjunction with other studies assessing messaging type, found that communication programs that focus on the individual benefit of vaccination are the most effective at increasing vaccine intention and uptake [58,59]. Indeed, attitudes such as perceived benefit and risk have been shown to be related to HPV vaccination acceptance [60,61]. Our findings are consistent with similar patterns observed in COVID-19 vaccination, with hesitancy linked to concerns about effectiveness, safety, and necessity [62]. Communication programs focusing on individual benefits can increase vaccine intention and uptake. Healthcare authorities can target hesitant parents through campaigns and endorsements, while using validated vaccine hesitancy scales to measure hesitancy consistently and develop targeted interventions [62].

The subgroup findings showed that there was a significant effect on parents’ confidence and attitude toward vaccination. For every one-unit increase in the scores for confidence and attitudes, the likelihood that parents would allow their adolescents to vaccinate increased by 28% and 21%, respectively. The 3R communication model significantly reduced the parents’ negative attitude toward vaccination acceptance. For every one-unit decrease in the negative attitudes score, the likelihood that the parents would allow their adolescent to vaccinate increased by 15%. The adolescents’ confidence and attitude scores toward vaccination significantly increased. The one-unit increase in the scores for confidence and attitude the likelihood that the adolescent would get vaccinated increased by 18% and 7%, respectively. However, the percentage increases in parents’ confidence and attitude scores were greater than those in adolescents. Previous studies found similar results regarding parents’ attitudes and knowledge about cervical cancer and HPV, as well as the subsequent intention to receive the vaccination. Perceptions from parents about the HPV vaccine correlated with adolescent uptake of vaccination [63]. In related studies conducted in the United States, Fontenot, Domush, and Zimet concluded that the importance of parents’ perception should be considered when creating messaging about HPV vaccine acceptance and uptake [64]. In a study conducted in Nigeria among female adolescents, Ndikom and Oboh found that adolescents’ perceptions were significantly associated with the uptake of the HPV vaccine, and parental approval for the HPV vaccine was also found to be significantly associated with vaccine uptake [65]. These studies provide evidence that parental and adolescent perceptions of the HPV vaccine must be addressed in HPV educational intervention programs to increase their intention of HPV vaccination acceptance.

Our community-based educational intervention among parents and their adolescents was created utilizing the 3R (reframe, reprioritize, and reform) communication model and was shown to significantly improve attitude, knowledge, confidence, and intention regarding HPV vaccination acceptance [59]. The significant increase in participants’ post-intervention scores may be attributed to the method of delivery of the intervention through utilizing the specific framework of the 3R model as performed in this study. The in-person group presentation, moderate group size, and method of communication delivery have been shown to be an effective agent of change to improve the knowledge and attitudes toward cervical cancer and HPV/HPV vaccination, along with the increased likelihood of intention to accept vaccination [66,67].

Among other demographics, previous studies with HPV-focused interventions delivered through various media forms have shown the effectiveness of educating parents [68,69,70,71,72,73]. This communication model may be used concurrently with other methods of educational delivery (e.g., online education and websites) for further reach and endorsement of the HPV vaccination. To the best of our knowledge, this research is one of the first presenting evidence of the effectiveness of the 3R communication model to improve the intention to receive HPV vaccination in the Ghanaian population.

This study had some limitations. The quasi-experimental design nature of this study did not include a control group to compare results and confirm any causal relationships between the intervention and results. Additionally, since we used a single group pre- and post-intervention design, there may have been confounding factors influencing the final results; therefore, the findings of the study should be interpreted with caution. Moreover, without long-term follow-up, this study could only determine the intent of HPV vaccination and not the actual uptake of vaccination. In future research, a randomized control trial with a longer follow-up time is necessary to analyze the results of actual uptake of vaccination and the long-term effectiveness of improving parents’ knowledge and attitudes toward the HPV vaccination. Secondly, this study design utilized a convenience sample of adults and children willing to participate with interest in the study topic. This limitation could be addressed in the future with additional resources by enlisting a comparison group of parents and adolescents who opted not to participate in the study. Additionally, this study did not evaluate actual HPV vaccination behavior as we had limited resources to assess whether the intention for HPV vaccination acceptance among parents and their unvaccinated adolescents resulted in actual vaccination. In future interventions, an arrangement could be made to have a mobile vaccination clinic available so that participants who are willing and able to afford it can have access to vaccination right away. Although the cost of a vaccine can be a barrier to vaccination, after the intervention, the participants did not seem to worry about their ability to pay. One of our survey questions asked participants the following: “If my adolescent gets the vaccination, I will not be able to pay the vaccination bill”; this item did not seem to have any impact on the participants’ self-confidence for HPV vaccination acceptance. However, future studies to access actual vaccination will help determine participants’ ability to afford it. The strengths of this study include being one of the first to test the effect of intervention on Ghanaian parents’ attitudes and behaviors toward HPV vaccination acceptance. Additionally, this study assessed the efficacy of the 3R communication model in increasing the HPV vaccination intent of Ghanaian parents and adolescents. This information has important implications for the importance of education and addressing attitudes and perceptions when informing parents and adolescents of HPV vaccination. According to previous studies, framing the HPV vaccine message within the context of sexual behavior is ineffective because it solidifies the stigma and religious objections associated with vaccination [73]. Our 3R model approach shifts the HPV vaccination narrative from sexual behavior to cancer prevention, and this study provides a future framework for intervention studies to promote HPV vaccination in Ghana.

## 5. Conclusions

An intervention utilizing the 3R communication model was effective in increasing parental and adolescent knowledge, attitudes, and confidence toward the HPV vaccination, as well as intention of HPV vaccination acceptance. These findings suggest that the 3R communication model, when used to guide intervention, has the potential to increase HPV vaccination uptake in Ghana. This information has important implications for the importance of education and addressing attitudes and perceptions when informing parents of HPV vaccination.

## Figures and Tables

**Figure 1 vaccines-11-00890-f001:**
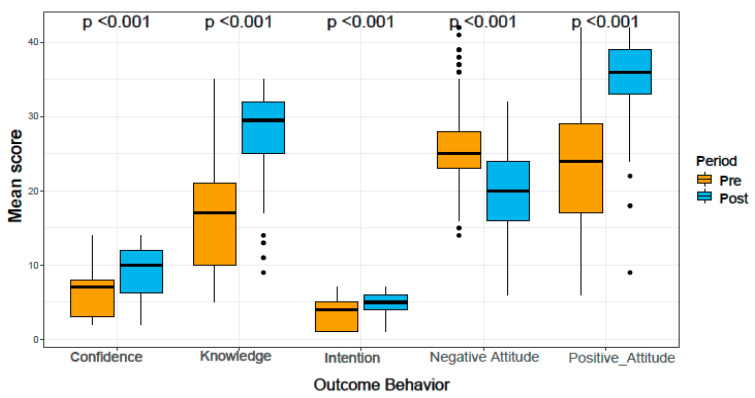
Boxplot showing changes in pre- and post-intervention score for confidence, knowledge, intention, negative and positive attitude (aggregated score for parents and adolescents).

**Table 1 vaccines-11-00890-t001:** Demographics of parents.

	Frequency (%)	Knowledge Mean (SD)	Confidence Mean (SD)	Attitude Mean (SD)	Intention Mean (SD)
Age					
Mean (SD) 46 (12.2)					
Sex					
Male	7 (8.2)	15.00 (3.14)	4.14 (1.2)	14.71 (2.94)	3.14 (0.74)
Female	78 (92.0)	10.51 (0.94)	2.58 (0.45)	10.79 (0.88)	1.36 (0.22)
Education					
Graduate degree or higher	5 (5.9)	0.00 (3.60)	4.00 (1.82)	−0.40 (3.28)	0.20 (0.88)
Undergraduate	5 (5.9)	12.40 (3.60)	1.80 (1.82)	13.60 (3.28)	1.00 (0.88)
High school	24 (28.0)	10.54 (1.64)	3.25 (0.83)	12.13 (1.50)	1.33 (0.40)
Less than high school	39 (46.0)	11.77 (1.29)	2.23 (0.65)	10.77 (1.78)	1.49 (0.32)
None	12 (14.0)	12.58 (2.33)	3.00 (1.17)	14.00 (2.11)	2.67 (0.57)
Marital status					
Married	52 (61.0)	10.50 (1.15)	2.63 (0.56)	11.37 (1.07)	1.56 (0.27)
Divorced	17 (20.0)	13.77 (2.02)	2.65 (0.98)	13.47 (1.88)	1.88 (0.47)
Never married	9 (11.0)	10.67 (2.77)	4.22 (1.35)	7.22 (2.58)	2.00 (0.65)
Remarried	7 (8.2)	7.00 (3.15)	1.43 (1.53)	8.57 (2.93)	−0.43 (0.74)
Insurance					
Yes	74 (87.0)	10.31 (0.96)	2.96 (0.46)	10.86 (0.91)	1.34 (0.23)
No	11 (13.0)	14.73 (2.49)	1.00 (1.20)	12.82 (2.36)	2.64 (0.59)
Employment					
Working	56 (66.0)	11.20 (1.12)	2.34 (0.53)	11.75 (1.04)	1.61 (0.27)
Not working	29 (34.0)	10.28 (1.56)	3.41 (0.74)	9.90 (1.45)	1.31 (0.37)
Child doctor					
Yes	5 (5.9)	8.60 (3.75)	0.80 (1.79)	6.00 (3.47)	−0.40 (0.88)
No	80 (94.1)	11.03 (0.94)	2.83 (0.45)	11.44 (0.87)	1.63 (0.22)
Income range					
>GHS 20,000	23 (27.0)	16.78 (1.59)	1.55 (0.83	15.83 (1.53)	2.61 (0.40)
>GHS 20,000–GHS 35,000	50 (59.0)	8.66 (1.07)	3.18 (0.57)	9.38 (1.04)	0.96 (0.27)
<GHS 35,000–GHS 50,000	12 (14.0)	8.833 (2.2)	2.92 (1.15)	9.33 (2.12)	1.67 (0.55)

**Table 2 vaccines-11-00890-t002:** Intervention score for outcome behavior by group and period.

	Pre-Intervention			Post-Intervention			Pre-Intervention	Post- Intervention
	Parent	Adolescent		Parent	Adolescent		Parent + Adolescent	Parent + Adolescent
Outcome Behavior	Mean (SD)	Mean (SD)	Est.	Mean (SD)	Mean (SD)	Est.	Mean (SD)	Mean (SD)
Confidence	6.71 (2.67)	5.64 (2.91)	1.07 *	9.41 (2.67)	9.41 (2.67)	0.089	6.17 (2.84)	8.96 (3.43)
Intention	3.86 (1.67)	2.72 (1.89)	1.14 **	5.36 (1.67)	4.09 (1.89)	1.27 **	3.29 (1.87)	4.73 (1.77)
Knowledge	17.40 (7.15)	15.72 (7.16)	1.68	28.28 (7.15)	28.58 (7.16)	−0.30	16.56 (7.19)	28.43 (5.13)
Negative attitude	25.35 (5.21)	26.85 (5.68)	−1.50	20.39 (5.21)	19.01 (5.68)	1.38	26.10 (5.48)	19.70 (5.43)
Positive attitude	25.38 (7.52)	21.46 (9.25)	3.92 **	36.49 (7.52)	34.42 (9.25)	2.07 *	23.42 (8.63)	35.46 (5.46)

* *p* < 0.05; ** *p* < 0.01.

**Table 3 vaccines-11-00890-t003:** Multinomial logistic regression model of intention to vaccinate.

	Parent ¹	Adolescent	Parent + Adolescent
	AdjOR (95% CI)	AdjOR (95% CI)	AdjOR (95% CI)
Confidence	1.28 (1.09, 1.51) **	1.18 (1.04, 1.34) **	1.19 (1.08, 1.30) **
Positive attitude	1.21 (1.03, 1.43) **	1.07 (1.00, 1.16)	1.09 (1.03, 1.17) **
Knowledge	1.12 (0.98, 1.31)	0.97 (0.88, 1.06)	0.98 (0.92, 1.06)
Negative attitude	1.15 (1.04, 1.28) **	1.01 (0.94, 1.09)	1.04 (0.98, 1.10)

^1^ Adjusted for baseline scores. ** *p* < 0.01.

## Data Availability

The data presented in this study are available on request from the corresponding author. The data are not publicly available due to participant’s privacy.

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
