# Peer review of "Assessing the Efficacy of the 3R (Reframe, Reprioritize, and Reform) Communication Model to Increase HPV Vaccinations Acceptance in Ghana: Community-Based Intervention"

_vaccines, 2023, doi:10.3390/vaccines11050890_

Round 1

Reviewer 1 Report

The document is well written and the research was carried out correctly. My only doubt is about the final sample analyzed (n = 85 + 85), regarding your statement on line 147 explaining that "a total sample size of 60 was used in this study".

Author Response

Find attached

Reviewer 2 Report

Several elements of the methods / early results sections need to be refined to improve the impact of this publication. The authors should note if the vaccine acceptance measure used is a validated tool and if so, provide citation to support that. Additionally, the TPB tool and scale used should be provided to interpret the numerical results presented in this paper.

- It is not clear if the participants were intentionally grouped in any way and how the participants interacted with each other during the face-to-face group sessions, which could influence outcomes in addition to the actual intervention delivered.

- In the beginning of the results, the authors indicate that the acceptance data was "recoded" without any description of what was recoded and how it could influence outcomes.

- It is not clear whether participants included in the analysis were required to participate in all 3 modules as both parent and child, which is important if they are assessing the impact of the entire approach. 

Author Response

Find attached

Reviewer 3 Report

introduction is lengthy and could be streamlined to enhance readability. Condensing some of the details on the 3R communication model, theory of planned behavior, and previous research findings would help focus the introduction on the key points and study objectives: needs to be summarized and parts maybe moved to the Discussion 

- in the Discussion about vaccine uptake as compared with childhood vaccination (lines 340-355): authors could contrast their findings with the similar patterns are observed in COVID-19 vaccination, with hesitancy linked to concerns about effectiveness, safety, and necessity. Communication programs focusing on individual benefits can increase vaccine intention and uptake. Healthcare authorities can target hesitant parents through campaigns and endorsements, while using validated vaccine hesitancy scales to measure hesitancy consistently and develop targeted interventions. (Suggested references: doi: 10.1016/j.vaccine.2021.02.039 , doi: 10.3389/fpubh.2021.752323 , https://doi.org/10.3390/healthcare11070972

Author Response

Find attached

Reviewer 4 Report

1.  These authors analyzed a communication model intervention to determine its impact on parents’ and adolescents’ HPV vaccination acceptability.  They met with the participants in a face-to-face interaction from 3 local churches in Ghana.  85 parents were involved and 85 adolescents were involved.  Mean scores for aptitude, knowledge, confidence, and intention for vaccine acceptance were increased with these interactions.  They suggest that the odds for HPV vaccination acceptability increased 22% for every 1 unit increase in the participants’ self-confidence and attitude scores.  Intention scores were higher for parents than adolescents.  They conclude that this model can provide a successful approach to improving HPV vaccination in this country.  This might have strong economic and medical effects since the rate of cervical cancer is relatively high in this country.
2.  Introduction: The introduction summarizes information regarding HPV in Ghana and the frequency of cervical cancer.  It then summarizes 3 R communication model.

3.  Methods: The methods are described in detail.
4.  Results: This communication model clearly resulted in definite improvements in the knowledge scores, confidence scores, and attention scores, and aptitude scores. A multinomial logistic regression model of intention to vaccination increased as confidence increased and as positive attitude increased in parents, in adolescents, and in the combined group.

5.  The discussion covers the project well.  Limitations discussed are appropriate.

Author Response

Find attached

Round 2

Reviewer 2 Report

Several of the changes cited in the response to the review do not actually appear in the manuscript. Namely:

- Participants included in the study participated in the entirety of all 3 sessions in sequence.

- The recoding clarification at the beginning of the results. 

- The limitations of the group format influencing potential outcomes and attitudes. 
